# Digital Health in Children’s Oral and Dental Health: An Overview and a Bibliometric Analysis

**DOI:** 10.3390/children9071039

**Published:** 2022-07-13

**Authors:** Peivand Bastani, Nithin Manchery, Mahnaz Samadbeik, Diep Hong Ha, Loc Giang Do

**Affiliations:** 1School of Dentistry, UQ Oral Health Centre, The University of Queensland, Brisbane, QLD 4006, Australia; p.bastani@uq.edu.au (P.B.); n.manchery@uq.edu.au (N.M.); d.ha@uq.edu.au (D.H.H.); 2Social Determinants of Health Research Center, Lorestan University of Medical Sciences, Khorramabad 6813833946, Iran; mahbeik@gmail.com

**Keywords:** digital, oral health, dental care, children, bibliometric analysis

## Abstract

Digital health technologies can widely increase access to oral health solutions and can make them easier to use and more accessible at all primary, secondary, and tertiary levels. This study aims to present a bibliometric analysis of published literature to identify the content, trends, and context of digital health technology use in children’s oral and dental health. After finalising the research question, the Scopus database was used to search systematically for related keywords from 1997 to 2022. The PRISMA methodology applied for systematic reviews was adopted to refine search results. VOS viewer software was applied to illustrate the topics and trends of digital health technology involved in children’s oral and dental health. An increase in use of the digital technologies was appeared in the index keywords after 2005. Computer-assisted therapy/surgery, computer simulation, computer program, image processing, nuclear magnetic resonance (NMR) imaging, and audio-visual equipment were more used index keywords in children’s dental care re-search from 2005–2015. Telemedicine, mobile application, virtual reality, and medical information were reported with the index keywords of dental caries, dental procedures, and dental anxiety after 2015. The study also identified a gap in the published literature in applying newer digital technologies, such as the Internet of Things (IoT) and gamification, in oral and dental health research and practice. There is a growing tendency to use digital technologies in children’s oral and dental health in recent years. Although the types and categorisations of the technology are typically diverse during the timeframe and by the area of dental services and oral health, identifying and categorizing these technologies based on oral health services could familiarise oral health policymakers with the application of the technology and help them design technology-based interventions to improve children’s oral health.

## 1. Introduction

Children’s oral health is a critical factor that can vastly affect their eating habits, speech development, sleeping patterns, overall development, well-being, and quality of life [1]. It is now well established that many physical, psychological, and social disorders are associated with poor oral health in children. Physical aspects such as low birth weight, preterm delivery and iron deficiency are linked to oral problems as well as psychological ones, and from a social perspective, the lost sleep, poor learning, poor self-esteem, and high school absence rate are related to poor pediatric oral and dental health [2]. Furthermore, different determinants strongly affect children’s access to oral health care services, including individual, social, cultural, and economic determinants, health policies, and availability of services [3]. Availability and access to oral and dental services as a main determinant should be mentioned by oral health policymakers and related interventions should be defined to achieve appropriate access and good oral health among children [3].

Applying digital technologies is among the new approaches that can broadly in-crease the availability and access to oral and dental care services [4]. Information and Communication Technologies (ICTs) and digital health technologies can widely increase access to oral health solutions; make them easier to use and more accessible at all primary, secondary, and tertiary levels; and also improve the effectiveness and moderate the cost of each approach [5]. At the same time, such these technologies have been applied for many purposes, from dental diagnosis and consultation to oral education and public awareness, as well as surgeries and treatment [6].

For children, the advantages of digital health technologies are even more distinguished, with increasing acceptance among parents and healthcare professionals. It can also help dental professionals to examine children in their most familiar atmosphere, such as their home environment, and save considerable time by not travelling to a clinic, thereby reducing the children’s and their parents/caregivers’ dental anxiety and fear [7]. Findings from a recent review indicated that applying teledentistry resulted in better management of pediatric patients [3]. Notably, the broad advantages of teledentistry also include providing patient/parents’ education, preventive care monitoring, post-treatment follow-up, and dental development assessment [8]. Along with all the advantages, applying digital health in pediatric dentistry thus potentially can reduce the oral healthcare inequalities which may be caused by the lack of access to specialists and timely oral and dental care services [9].

There are different studies aiming to conduct and compare a variety of digital technology-based interventions’ outcomes on children’s oral health. For instance, Zolfaghari et al. indicated that after a month of applying a simple app without gamification and its gamified version, the studied mothers’ knowledge and practice of oral health improved [10]. Wallace et al. demonstrated that after implementing a telephone consultation for children and their parents, both the number of unnecessary face-to face appointments and the waiting lists were decreased [11]. Hammersmith et al. have also shown that applying teledentistry in a pilot of a children`s hospital system led to positive attitudes of caregivers and dentists [12].

Hence, many oral health interventions/applications are discussed in this multi-disciplinary area comparing the outcomes of digital technology use in clinical examinations, consultations, and treatments with the traditional method for the children’s oral health conditions. This extensive influence of technologies on children’s oral and dental health highlights a need to understand contexts and contents of the topic ad-dressed in the current literature. In other words, what is known is that despite the influence of digital health technologies on children’s oral health, the general picture of research on the contexts and contents of the topic remains unknown. Therefore, it would be useful for oral health policymakers, oral and dental care providers, and pediatric dentists to firstly identify the types of applications/interventions based on digital health and secondly explore the characteristics and variety of digital technology interventions in children’s oral and dental health practice. Such new knowledge can help health policymakers to better identify the suitability of digitally based applications/interventions in children’s oral and dental health for future planning. Thus, this study aimed to present a bibliometric analysis of digital health interventions/applications among the published literature of children’s oral and dental health in order to identify research gaps and trends in the literature.

## 2. Materials and Methods

We used descriptive statistics to analyse the publication geography of the topic. In order to conduct the present bibliometric analysis and cluster digital health interventions’ topics and trends, VOS viewer software version 1.6.15 was used [13]. Firstly, a search strategy for the Scopus database was prepared to retrieve peer-reviewed, published articles in the area of “digital health interventions in children oral and dental health” (Table 1). The Scopus database was selected because Scopus covers more titles and research papers than the other popular biomedical databases [13]. As digital health is a new and emerging concept of technology, we specified the publication year 1996 as the start year of included documents for the Scopus search. The Preferred Reporting Items for Systematic Reviews and Meta-Analyses (PRISMA) [14] methodology was applied for refining the initial search results. A PRISMA flowchart was used to select the relevant articles from Scopus databases. The PRISMA statement of a flow diagram to help authors improve systematic reviews and meta-analyses can be found on http://www.prisma-statement.org (accessed on 20 May 2022) [14].

The results of the Scopus search were exported in the CVS file format, including all data elements such as citation information, bibliometrics, abstracts, and keywords. The electronic search was conducted in April 2022. In order to visualise the data, the information was imported to the VOS viewer software. The software aids in visualising all author keywords and index keywords related to “digital health intervention” and “children oral and dental health” through co-occurrence analysis, applying the full counting method. A thesaurus file was needed to demonstrate the meaningful concept mapping to perform data cleaning and terms merging. We generated the thesaurus file by entering these key words, as provided in the instruction manual of VOS viewer, into a simple text editor (Notepad++ 7.9.2 for Windows). This thesaurus file was finally uploaded into VOS viewer to replace and ignore specified terms for creating maps. This thesaurus file was used to merge terms, such as singular and plural (e.g., “mobile applications” and “mobile applications”; “cross-sectional study” and “cross-sectional studies”), and synonyms (e.g., “newborn” and “infant”; “mobile phone” and “cell phone”; “computer network” and “computer communication networks”). We ignored general terms that did not add value to the study, such as methodology-related terms, priority journal, questionnaire, article, human, countries. Descriptive statistics were used to report the publication geography of the topic. To make the maps simple and decrease term diversity, the threshold of the minimum number of occurrences of a keyword was set to 2. Thus, terms with fewer than 2 occurrences were not illustrated on the map. As digital health is a relatively new, emerging field of study, we set this threshold for better visualization of keywords. The threshold mentioned above is a desirable number to cancel keywords with misspelling and meaningless keywords.

Keywords network analysis was also applied to demonstrate the most researched keywords related to this topic and the relationship. VOS viewer was also applied to imply international collaboration via country co-authorship analysis. Network visualization maps were prepared to visualize the associations among the keywords. In these maps, nodes and labels were applied to show the items, which were then classified in clusters. The weight of each item here was a determinant for the size of the nodes and the item’s label. In other words, by increasing the weight of each item, the label and the denoting circle for that item increased. The weight of each item itself was identified by the links and the total link strength attributes. The links imply the number of each item’s links with other items, and similarly, the total number of links shows the overall strength of the links of an item with other items [15]. Different colours were used to represent the clusters to which a node has been allocated. The clustering method as a technique for categorizing items by their similarity is described by Waltman et al. [16,17]. Along with network maps, overlay maps were used to show a density visualization. Such a visualization can provide a quick overview of the main areas in a bibliometric network [18]. We have used overlay maps here to demonstrate the development of digital health over time in the included literature.

## 3. Results

In our work, we adapt the flow diagram to suit our needs for refining the obtained search results (Figure 1).

We retrieved 374 documents by entering the search query string of TITLE-ABS-KEY (“digital health interventions in children oral and dental health”) into the Scopus search engine on 10 April 2022. As digital health is a new and emerging concept of technology, we specified the publication year 1996 as the start year of included documents for the Scopus search. Finally, 80 documents were subjected to analysis. The list of included articles is available at Appendix A.

### 3.1. Descriptive Analysis of the Publications

#### 3.1.1. Country Analysis/Geographical Distribution

For distribution of articles according to the countries of publication, most that were published were from the United Kingdom (12/80; 15%), followed by the United States of America (USA) (10/80; 12.5%) (Table 2). Thirty-three countries met or exceeded the threshold of one (minimum number of one document for a country) and appeared on the map. Collaborations were identified among nine countries in three main clusters. The first cluster includes Germany, Japan, Singapore, and Switzerland, while the second cluster consists of Australia, Canada, and the USA. Denmark and Sweden were situated in the third cluster of collaborations among countries in these publications. Table 2 shows countries with the most recent average contributions to this research field.

According to the year of publication, most of the articles were published in 2021 (16; 20%). The second concentration of published articles then occurred in 2020 (11; 13.75%) and 2017 (11; 13.75%). Figure 2 shows the trend of publication in the area of digital health applications/interventions in children’s oral and dental health. According to Figure 2, a sharp increasing trend of publications is revealed after 2015.

#### 3.1.2. Research Topics/Co-Occurrence of Keywords

Scopus determined the index keywords and the author keywords. In comparison to the index keyword, the author keyword co-occurrence analysis revealed a smaller number of keywords (keyword display = 32) (Figure 3).

Different patterns of author keyword co-occurrence were obvious in seven clusters based on the 80 records by VOSviewer (Figure 3) (details of the 80 articles are presented in Appendix A). For instance, tele-dentistry and apps were applied in the COVID-19, dental trauma, and emergency management areas for pediatric dentistry by researchers.

Telemedicine, in general, was also used along with mother’s knowledge in oral public health studies. Similarly, pediatric dental concepts were mentioned with some digital sources, such as databases.

Figure 3 illustrates in light green colour the more sophisticated and newer digital technology interventions, such as virtual reality (VR), which was mainly used for pain control, oral disease prevention, and dental health practices. It was seen that autism patients were an important group for using this kind of modern technology among the included studies.

The digital cell phone was mainly applied as a technology intervention for dental caries, dental plaque, and oral health education. Caries risks assessment and Cariogram were also being used as an app by the digital cell phone (illustrated in dark green colour in Figure 3).

Finally, the Internet and social media were applied for improving health-related behaviors, such as oral hygiene and tooth brushing.

From the novelty aspect of digital technologies in oral and dental health (Figure 4), social media and telemedicine were frequently author keywords used with public health programs, mother’s knowledge, COVID-19, prevention, and autism in 2020. Apps, cell phones, the internet and tele-dentistry were author keywords used more with oral hygiene, dental pediatrics, oral health education and promotion, dental trauma, and tooth brushing during 2015 to 2020 in the publications.

An increase in use of the digital technologies was appeared in the index keywords after 2005 (Figure 5). Computer-assisted therapy/surgery, computer simulation, computer program, image processing, nuclear magnetic resonance (NMR) imaging and audio-visual equipment were more used index keywords in children’s dental care research between 2005–2015. Telemedicine, mobile application, virtual reality, and medical information were reported with the index keywords of dental caries, dental procedures, and dental anxiety after 2015.

Alternatively, and in contrast with author keywords, the index keyword co-occurrence showed 174 keywords (Figure 6). It can be seen that the dominant index keywords were female, male, preschool child, oral hygiene, dental caries, and Internet. Regarding the frequency of occurrence of keywords, two digital technologies, the Internet and cell phone, had the greatest occurrence among the research topics with 15 and 10, respectively. Furthermore, the most frequent oral and dental-health-related terms were oral hygiene and dental caries, with 21 and 17 occurrences among the literature. From the age group aspect, preschool occurred most frequently in articles. This means that these keywords were the most discussed by researchers in the January 1997–April 2022 period. Furthermore, from the study methodology perspective, two methods (clinical trial/randomized and cross-sectional) have the most occurrences, 27 and 13 times, respectively, among the publications. Based on this result, it seems that the co-occurrence analysis using index keywords produced a clearer understanding of the current state of the topic in comparison to author keywords co-occurrence analysis.

As illustrated in Figure 6, the index keywords related to different clusters were shown in different colours in the network visualization map. Each keyword is shown by a node in a colour corresponding to its cluster. Keywords occurring together in research are indicated in the cluster. The thickness of the connecting line shows the strength of pairs of keywords. Seven clusters were identified that all were algorithmically grouped to show those index keywords with close relationships. For better illustration and description of the results related to each cluster, the keywords were displayed with cluster colours and tabulated based on related keywords in each cluster in Table 3.

From the first cluster (in red) (Table 3), three main relations are clear: firstly, the relationship among digital technologies, including web browser, computer communication, dental records, telemedicine, and the Internet with the oral health and related problems like tooth eruption, ambulatory care, deciduous teeth, dental enamel hypoplasia, dental restoration, and pulpectomy; secondly, the relationship explored among these digital technologies with the category of communication-related words such as distance education, rural population, interpersonal communication, interprofessional relationships, and public relations; lastly, the relation revealed among the mentioned digital technology keywords with health-service-related items such as appointment and scheduling, health centres, hospital management, health personnel attitudes, insurance, organization and management, and reimbursement. Notably, reviewing the red cluster keywords also indicated that digital interventions situated in this cluster have mainly been retrieved in articles with qualitative and pilot studies.

Cluster 2 (in green) shows the variety of digital technology interventions that are associated with children’s oral health. The most notable among them was computer assisted systems, including audio-visual equipment, computer assisted diagnosis, computer assisted surgery, computer assisted therapy, computer assisted tomography, computer program, computer simulation, image processing, user-computer interface, three-dimensional imaging treatment, computer-assisted, and nuclear magnetic resonance imaging. The later variety of digital technologies was accompanied with oral and dental-health-related items or keywords such as cephalometry, dental aesthetics, malocclusion, maxillofacial surgery, and orthodontics. Another relationship among the digital technology interventions observed in this cluster with other index keywords was seen with treatment planning and outcomes, psychological terminology like self-concept and patient satisfaction, and radiography and mothers. This digital applications/intervention cluster was mainly retrieved from the methodological approach of case reports.

Thirdly, digital technology applications/interventions, including social media, medical information, and tablets, were used with various oral and dental health concepts, and were presented in blue, including anticaries agent, cariostatic agent, cleft lip, cleft palate, dental caries, gingiva disease, and incisor teeth. Other oral and public health issues that were indexed in this cluster include: dmft index, fluorides, sugar intake, toothpaste, dietary source, and supplements. From the methodological approach, the digital technology interventions mentioned in this cluster were mostly associated with cohort and cross-sectional studies, comparative studies, clinical controlled trials (randomized clinical trials), and epidemiologic methods.

As shown in Table 3, only two digital technology applications/interventions (diagnostic image and databases) are located in the fourth cluster (in purple). These interventions include a combination, with a variety of oral health and dental issues such as tooth loss, tooth injury, tooth disease, tooth crown, tooth avulsion, jaw, periodontal disease, dental photography, and follow-up. This digital technology cluster has been used in the publications aiming at different age groups, from preschool children, young adults, and middle-aged to older age groups. Other related terminology with diagnostic image and databases in this cluster includes health knowledge, attitude and practice, quality of life, risk assessment, and prognosis. Retrospective studies and follow-up studies were among the main methodological approaches combined with the mentioned digital technology interventions in this cluster.

In cluster 5 (in yellow), three groups of digital applications/interventions were illustrated, including computer assisted therapy, virtual reality, and visual analogue scale. Notably, dental anaesthesia, dental anxiety, dental care, dental procedure, pain measurement, nociception, and tooth extraction were among oral and dental-health-related terminologies among school children and adolescents that were combined with the digital applications/interventions in this cluster. In this cluster, the main methodological approaches were from randomized clinical trials, other major clinical studies, and crossover studies. From the health service approach, the present technologies were also related to preventive health services.

The sixth cluster (in orange) has mainly used three types of general digital technologies containing software, program evaluation, and information processing in combination with oral and public health concepts such as dental and dental health education, general and dental practice, epidemiology, health evaluation and promotion, healthcare quality, mouth and oral hygiene, perception, teaching, and skills. Another significant keyword index in this cluster is COVID-19, which indicates the combination of general digital technologies and oral and public health during the pandemic era.

Finally, the last cluster (in brown) mainly mapped three other digital technology interventions, including cell phone, mobile application, and video recording. This combination of technologies was retrieved in prospective studies and accompanied by the more behavioral aspect of oral and dental health, such as child behavior, dental plaque index, tooth plaque, and tooth brushing.

## 4. Discussion

This bibliometric analysis revealed seven clusters of digital technologies in children’s oral and dental health in 80 included articles during 1997–2022. An increase in use of the digital technologies appeared in the index keywords after 2005. Computer-assisted therapy/surgery, computer simulation, computer program, image processing, nuclear magnetic resonance (NMR) imaging and audio-visual equipment were more used index keywords in children’s dental care research from 2005–2015. Other digital technologies, such as telemedicine, mobile application, virtual reality, and medical information, occurred with the terms of dental caries, dental procedures, and dental anxiety after 2015. At the same time, although a sharp increasing trend of publications is revealed after 2015, most of the articles were published in 2021, which can indicate the simultaneous occurrence of COVID-19 and development of teledentistry. The study also identifies a gap among the publications in applying some of the newer digital technologies, such as IoT, artificial intelligence, and gamification (online gaming, video games, etc.) in oral and dental health.

According to the present bibliometric analysis, one of the new co-occurrences of applying digital technologies in children’s oral and dental health has focused on COVID-19, autism, child/parents, and child/teachers’ relationships. These new aspects of applying digital technology in the area of children’s oral and dental health open new avenues for dental health researchers as well as oral health policy makers who use the outcomes of the research for decision making not only to distinguish novel use of technology in the area but also to try to design applied applications/interventions to increase the access of the population to the related digital technologies and oral and dental health services, and as a result improve the general oral health among the children`s population. During and after the COVID-19 era, as Javaid et al. emphasized, a change may have happened in the use of digital technology in dentistry [19]. A scoping review also highlights a shift both in the teledentistry landscape and future clinical and research opportunities. It also highlights the way rural and remote regions during, and post COVID-19 era are benefiting from teledentristy though increased access and utilisation. [20].

Applying virtual reality is also among other modern technologies used for both dental education at dentistry schools and improving oral health literacy among the mothers, schoolteachers, school children and adolescents, particularly during and after the COVID-19 pandemic [21]. According to the results of a systematic review, applying virtual reality in dental education has increased rapidly during the COVID-19 era and could improve the students’ access to more useful theoretical content and clinical dental expertise remotely [22]. Despite any similar bibliometric analysis existing in this area, the results of a bibliometric analysis on the impacts of COVID-19 on dentistry have confirmed the considerable increase in global academic publications in Scopus related to the impacts of COVID-19 in dentistry [23]. These impacts are more notable in the scope of dental education, prevention, and practice as well as technology application [24].

On the aspect of oral and dental health education and literacy, other results of the study have focused on the co-occurrence between social media and dietary intake and type, sugar and fluorides consumption, and the child–parent relationship. Similarly, digital technologies such as cell-phone-via-mobile-application and videorecording demonstrated a close co-occurrence between children’s oral health behaviours. This evidence emphasized restructuring oral and dental health services and care by applying digital technology in the current century. Such a paradigm shift and restructuring of dentistry among publications demonstrates a new period of patient-oriented, technology- based, prevention-based, and outcome-centred oral and dental health service delivery for children [25] that can shed light for researchers on new interventions. These new interventions can enable dentists and oral healthcare workers to apply modern technologies and at the same time empower the whole community—particularly families, mothers, and teachers—to accept digital technologies and use them for the purpose of children`s oral and dental health.

Telemedicine and teledentistry are among the digital technologies in the area of pediatric dentistry that should be considered from different aspects according to the present results. While these kinds of technologies show a co-occurrence with some oral and dental procedures, such as dental restorations, tooth eruption, deciduous tooth, pulpectomy, tooth radiography, and enamel hypoplasia in recent publications, some other issues are also observed that need new attention from a different perspective. For instance, the co-occurrence among telemedicine and terms like insurance and reimbursement opens new windows for health service managers and policymakers to find mechanisms for defining new health basic benefit packages that include telemedicine and teledentistry as acceptable procedures from a third-party insurer perspective and reimbursement to dentists and insurance coverage, particularly for rural populations and distant areas. Clear policy statements are necessary in this area to both facilitate applying the technology and increase the access of the children and their families to dental services and at the same time create incentives for dentists and specialists to apply these technologies. Sufficient attention to the concepts of feasibility evaluation of teledentistry, acceptability, and sustainability of the technology [12] are among other important considerations that should be mentioned by health service researchers and oral health policy makers. At the same time, according to the results of Waqas et al., despite a constant increase in applying telemedicine and teledentistry in the publications, it is notable that most of these articles are conducted in high-income developed contexts [26]. This can open a new window of consideration for global oral health policymakers in the aspect of equality in access to the technology and decreasing the digital divide.

As it is not mandatory and common to appraise the content of the articles to determine the eligibility of the included articles, it is probable that some articles with non-intervention approaches were included in the bibliographic analysis, even though reviews, letters, notes, editorials, and conference papers were excluded. A comprehensive descriptive and bibliometric analysis that presents a synthesised overview of the topic is among the strengths of the study.

## 5. Conclusions

This bibliometric study has identified a growing tendency to use digital technologies in children’s oral and dental health. Although the types and categorizations of the technology are typically diverse during the timeframe and according to the area of dental services and oral health, it can be a great opportunity for policymakers to facilitate access to dental services, the population’s literacy, dental education, and oral and dental service delivery by applying appropriate technologies. This bibliometric study, along with health technology assessment and policy studies, can help policymakers identify the trend, type, application, and pros and cons of technology application in oral and dental health for the vulnerable group of children and pave the way for using more effective and suitable technologies for access improvement and improved children’s oral health according to the contextual determinants.

## Figures and Tables

**Figure 1 children-09-01039-f001:**
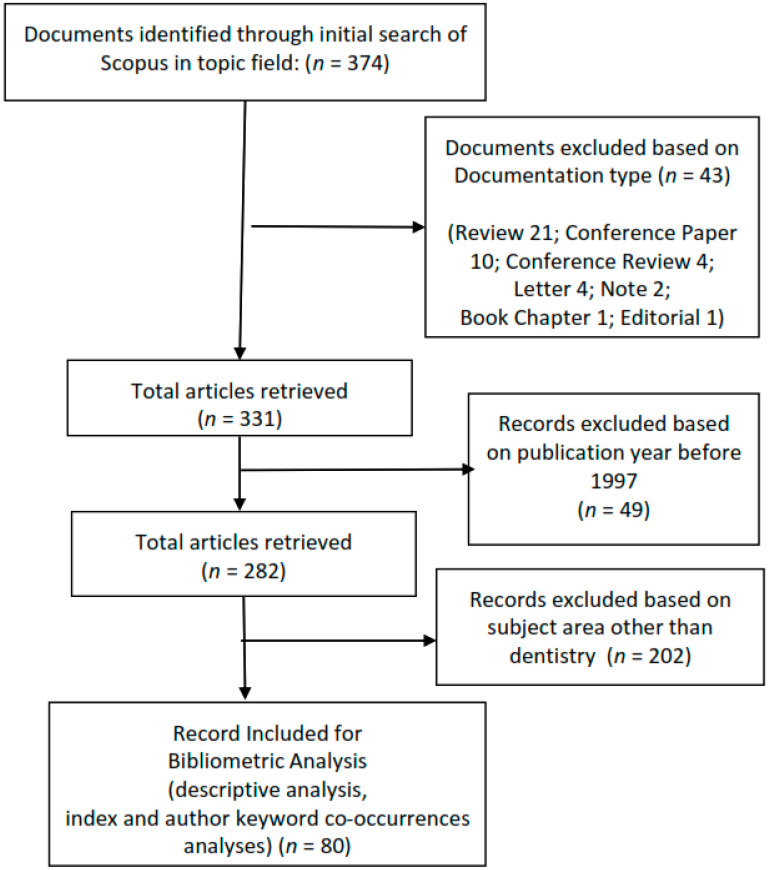
PRISMA flow charts of procedures used in the identification of final article for the bibliometric review.

**Figure 2 children-09-01039-f002:**
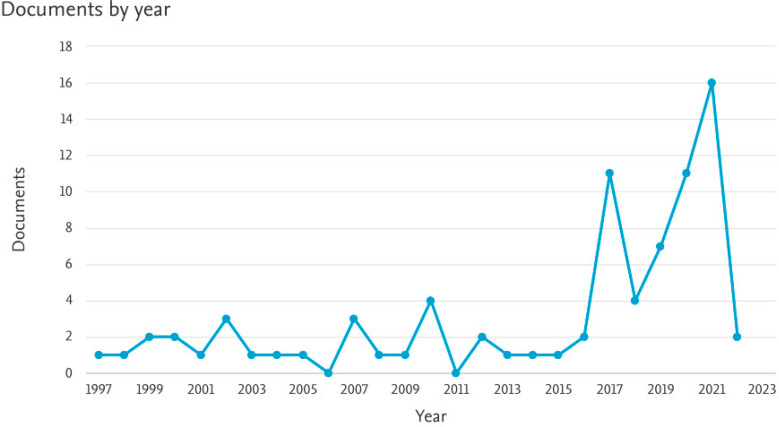
Trend of publication during the timeframe (Scopus source).

**Figure 3 children-09-01039-f003:**
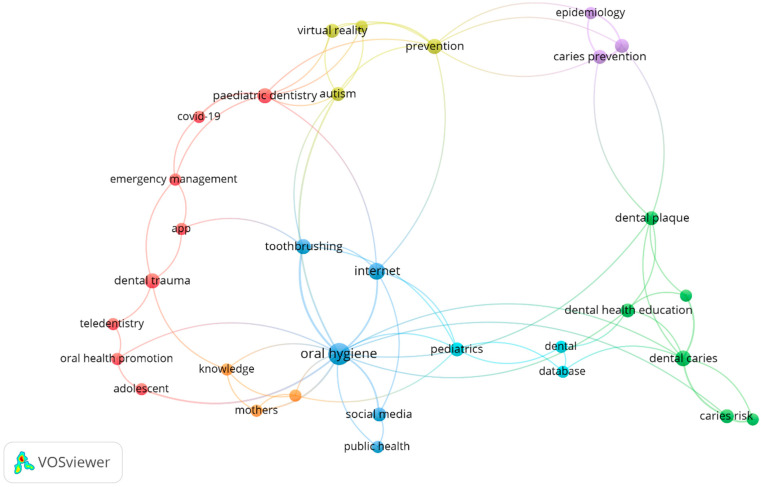
Network visualization map of the author keywords (keyword threshold = 2; displays 32 out of 211 keywords). Each colour represents different patterns of co-occurrences based on multiple keywords retrieved from the dataset.

**Figure 4 children-09-01039-f004:**
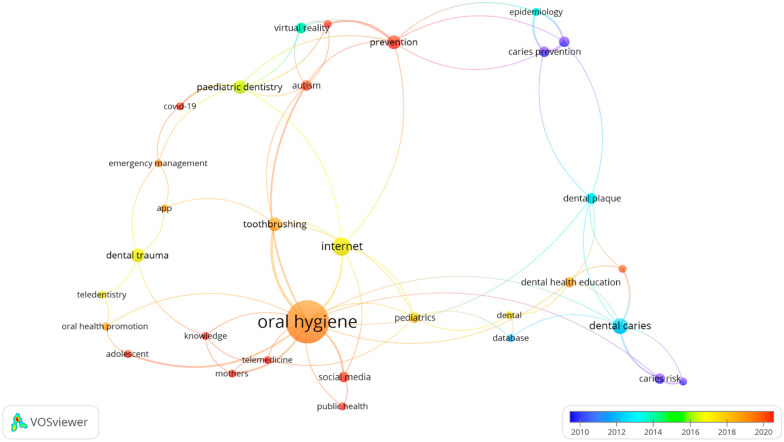
Overlay visualization map of the author keywords (keyword threshold = 2; displays 32 out of 211 keywords). Circles in dark and light blue and green represent those oral and dental related topics and digital technologies with an older average publication year. Circles in yellow, orange, and red demonstrate the terminologies with newer and more recent average publication year.

**Figure 5 children-09-01039-f005:**
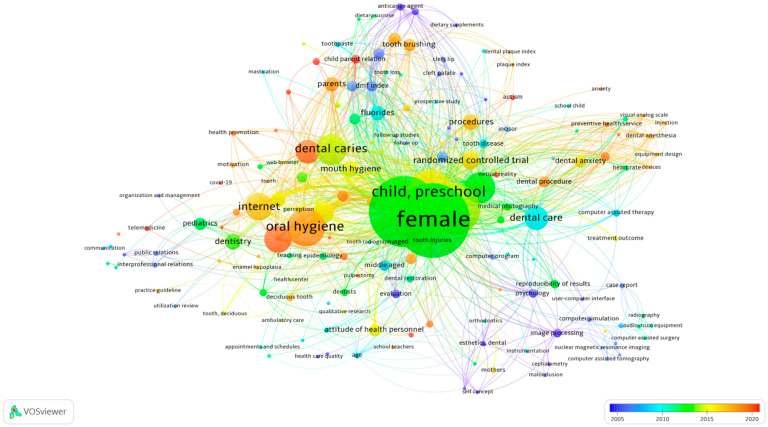
Overlay visualization map of the index keywords (keyword threshold = 2; displays 174 out of 559 keywords). Circles in dark and light blue and green represent those oral and dental related topics and digital technologies with an older average publication year. Circles in yellow, orange, and red demonstrate the terminologies with newer and more recent average publication year.

**Figure 6 children-09-01039-f006:**
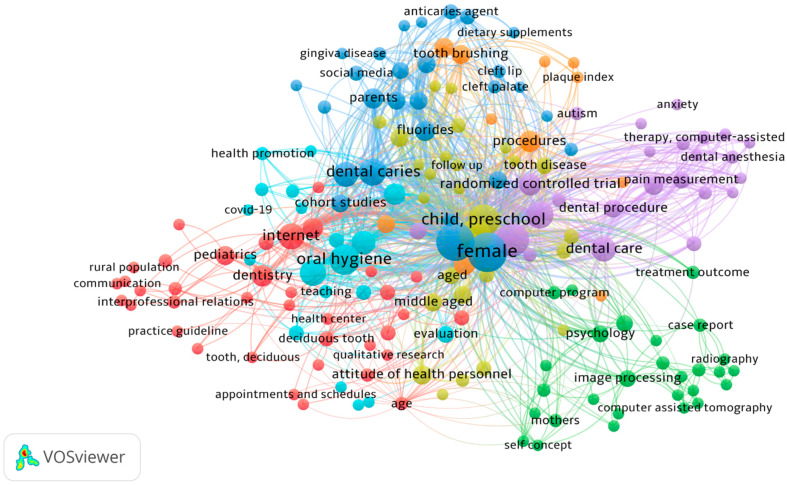
Network visualization map of the index keywords (keyword threshold = 2; display 174 out of 559 keywords). Each colour represents different patterns of co-occurrences based on multiple keywords retrieved from the dataset.

**Table 1 children-09-01039-t001:** Search strategy.

NO.	Construct	Search Field/Limits
#1	“Oral health” OR (Oral AND Health) OR (Diagnosis AND Oral) OR “Mouth Diseases” OR “Dental Health” OR “Dental care” OR (Dental AND Health) OR “Oral and dental health” OR “teeth health” OR “tooth health”	In: Topic (Title, Abstract, Keywords)
#2	“digital technology” OR “digital tool” OR “digital health” OR telemedicine OR “mobile health” OR mHealth OR eHealth OR “artificial intelligence” OR app OR smartphone OR software OR “information technology” OR electronics OR computer or internet OR online OR “Information and Communication Technology” OR Computers OR tablet OR phone OR iPad OR “mobile technology” OR “Cellular phone” OR “cellular phones” OR telecommunications OR “mobile applications” OR web-based OR “mobile apps” OR “text message” OR SMS OR “short message Service” OR “portable game” OR handheld OR PDA OR “personal digital assistant” OR “social media” OR “online social networking” OR “virtual reality” OR avatars OR “online gaming” OR “video games”	In: Topic (Title, Abstract, Keywords)
#3	paediatric OR child OR children OR Childhood OR infant OR toddler OR newborn OR preschool * OR pre-school * OR newborn OR “school age”	In: Topic (Title, Abstract, Keywords)
#4	#1 AND #2 AND #3	Language: English Publication year > 1996 Subject area: Dentistry Documentation type: Article

* represents any group of characters. * behind the root word and the search will look for all variations of the root word.

**Table 2 children-09-01039-t002:** Distribution of the articles according to the number of contributions of the countries.

Country	Number of Publications
United Kingdom	12
United States	10
India	8
Brazil	6
Germany	6
Others	36 (80 total)

**Table 3 children-09-01039-t003:** Clusters of index keywords related to digital health in children’s oral and dental health.

Cluster	Colour	Number of Keywords	Digital Technology Keywords	Selected Keywords
1		37	Computer communication, Electronic dental records, Internet, Web browser, Telemedicine	Interpersonal communication, dental restorations, tooth eruption, deciduous tooth, pulpectomy, tooth radiography, rural population, enamel hypoplasia clinical competence, distance education, insurance, reimbursement.
2		28	Audio visual equipment, Computer assisted diagnosis, Computer assisted surgery, Computer assisted therapy, Computer assisted tomography, Computer program, Computer simulation, Image processing, User-computer interface, Three-dimensional imaging, Nuclear magnetic resonance imaging	Treatment planning, treatment outcome, surgery, cephalometry, esthetics dentistry, instrumentation, malocclusion, maxillofacial surgery, orthodontics, patient satisfaction, psychology, mothers, self-concepts
3		28	Social media, Medical information, Tablet	Cohort studies, comparative studies, controlled clinical trial, cross-sectional studies, epidemiological method, health survey, cariostatic agent, cleft lip, cleft palate, dental caries, dmft index, fluorides, incisor, sugar intake, toothpaste, child parent relation, dietary source, dietary supplement
4		27	Diagnostic image, Database,	factual, medical photography Attitude of health personnel, child, pre-school, health knowledge, attitude, periodontal disease, risk assessment Tooth avulsion, tooth crown, tooth disease, tooth injuries, tooth loss, retrospective study
5		24	Computer assisted therapy, Virtual reality, Visual analogue scale,	Devices, adolescent, anxiety, autism, dental anaesthesia, dental anxiety, dental care, dental procedure, equipment design, pain management, preventive health service, school child, therapy, tooth extraction randomized clinical trials, cross over studies
6		19	Software, Program evaluation, Information processing,	Teaching, oral hygiene, mouth hygiene, health promotion, health care quality, general practice, dental health Dental education, perception, motivation, COVID-19, Epidemiology
7		11	Cell phone, Mobile application, Videorecording	Tooth brushing, tooth plaque, child behaviour, procedure, dental plaque index Prospective study

## Data Availability

Not applicable.

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
