# Peer review of "Digital Health in Children’s Oral and Dental Health: An Overview and a Bibliometric Analysis"

_children, 2022, doi:10.3390/children9071039_

Round 1

Reviewer 1 Report

We read with great interest the manuscript with title “Digital Health in Children’s Oral and Dental Health: An Overview and A Bibliometric Analysis” aiming to present a bibliometric analysis of published literature to identify the content, trends, and context of digital health technology use in children`s oral and dental health.

The text of the manuscript is interesting, and the work is well done, though not entirely consistent with the systematic review methodology. However, the subject matter is interesting, the way the results are presented is captivating and certainly increases the readability of the text itself.

Minor corrections should be made and please check the spelling of the text and the style carefully as there are small mistakes.

Please find hereafter the comments that need to be addressed before resubmission.

Introduction 

Lines 58-59 “Findings from a recent review indicated that applying tele-dentistry resulted in better management of pediatric patients. “Please add citation

Lines 70-73 “Wallace et al. (2021) demonstrated that after implementing a telephone consultation for children and their parents, both the number of unnecessary face-to face appointments and the waiting lists were decreased [11]. Hammer-smith et al. (2021) have also shown that applying tele-dentistry in a pilot of children`s hospital system led to positive attitudes of caregivers and dentists [12]“. Please delete the year of publication in the text in parentheses. If the year of publication is information that the authors consider important, enter "the study by xy published in ...., showed that…" or similar.

Results

Lines 143-145: “PRISMA flowchart was used to select the relevant articles from Scopus databases.  (Figure- 1). The PRISMA statement found on http://www.prisma-statement.org of a flow diagram to help authors improve systematic reviews and meta-analyses [15]. In our work, we adapt the flow diagram to suit our needs for refining the obtained search results” This part should be moved to Material and Methods section.

Lines 150-151: “As digital health is a new and emerging concept of technology, we specified the publication year after 1996 as the start year of included documents for the Scopus search” also, this is methodology description.

Lines 180-199. “Different patterns of keyword co-occurrence were obvious in seven clusters (Figure 3). For instance, tele-dentistry and apps were applied in the COVID-19, dental trauma, 181 and emergency management areas for pediatric dentistry by researchers. 

Telemedicine, in general, was also used along with mother`s knowledge in oral pub- 183 lic health studies. Similarly, pediatrics dental concepts were mentioned with some digital 184 sources like databases. 

The more sophisticated and newer digital technology interventions such as virtual reality (VR) was mainly used for pain control, oral disease prevention and dental health  practices. It was seen that autism patients were an important group for using this kind of 188 modern technologies among the included studies. 

Digital cell phone was mainly applied as a technology intervention for dental caries, 190 dental plaque, and oral health education. Caries risks assessment and Cariogram were 191 also being used as an app by digital cell phone. 

Finally, the Internet and social media were applied for improving health-related be- 193 haviors, such as oral hygiene, and tooth brushing.  From the novelty aspect of digital technologies in oral and dental health (Figure 4),  social media, telemedicine and virtual reality were frequently used with public health programs, mother`s knowledge, COVID-19, prevention, and autism in 2020. While apps,  cell phones, the internet and tele-dentistry were used more with oral hygiene, dental pediatrics, oral health education and promotion, dental trauma and tooth brushing during  2015 to 2020 in the publications. “ Please add citations of the articles cited in this part.

Reviewer 2 Report

This study provides new insights into the published literature on digital health technology use in children's oral and dental health. The text is well-written, figures and tables are informative, and the methods are sound. I have minor comments on the paper that are largely designed to improve the interpretation of the manuscript.

Minor comments:

The authors showed that there was an increase in the use of sophisticated digital technologies in publications after 2005 (abstract and discussion). However, there are no data in the results section to support that inference. Please, include these data in the results section.

The quality of the figures is low, especially the network plots, making it difficult to read the keywords.

Some excerpts from the results section are actually discussion, such as “The more sophisticated and newer digital technology interventions such as virtual reality was mainly used for pain control, oral disease prevention, and dental health practices. It was seen that autism patients were an important group for using this kind of modern technology in the included studies. Please, review the Results section.

Reviewer 3 Report

I suggest moore similar studies cited in the Discussion chapter, for comparison with the results of the present study. 
